

# *dxpr*: an R package for generating analysis-ready data from electronic health records—diagnoses and procedures

Yi-Ju Tseng[1,2], Hsiang-Ju Chiu[3] and Chun Ju Chen[3,4]

[1] Department of Information Management, National Central University, Taoyuan, Taiwan
[2] Department of Laboratory Medicine, Chang Gung Memorial Hospital at Linkou, Taoyuan, Taiwan
[3] Department of Information Management, Chang Gung University, Taoyuan, Taiwan
[4] Department of Information Management, National Taiwan University, Taipei, Taiwan

## ABSTRACT

**Background**. Enriched electronic health records (EHRs) contain crucial information related to disease progression, and this information can help with decision-making in the health care field. Data analytics in health care is deemed as one of the essential processes that help accelerate the progress of clinical research. However, processing and analyzing EHR data are common bottlenecks in health care data analytics.

**Methods**. The *dxpr* R package provides mechanisms for integration, wrangling, and visualization of clinical data, including diagnosis and procedure records. First, the *dxpr* package helps users transform International Classification of Diseases (ICD) codes to a uniform format. After code format transformation, the *dxpr* package supports four strategies for grouping clinical diagnostic data. For clinical procedure data, two grouping methods can be chosen. After EHRs are integrated, users can employ a set of flexible built-in querying functions for dividing data into case and control groups by using specified criteria and splitting the data into before and after an event based on the record date. Subsequently, the structure of integrated long data can be converted into wide, analysis-ready data that are suitable for statistical analysis and visualization.

**Results**. We conducted comorbidity data processes based on a cohort of newborns from Medical Information Mart for Intensive Care-III (n = 7,833) by using the *dxpr* package. We first defined patent ductus arteriosus (PDA) cases as patients who had at least one PDA diagnosis (ICD, Ninth Revision, Clinical Modification [ICD-9-CM] 7470*). Controls were defined as patients who never had PDA diagnosis. In total, 381 and 7,452 patients with and without PDA, respectively, were included in our study population. Then, we grouped the diagnoses into defined comorbidities. Finally, we observed a statistically significant difference in 8 of the 16 comorbidities among patients with and without PDA, including fluid and electrolyte disorders, valvular disease, and others.

**Conclusions**. This *dxpr* package helps clinical data analysts address the common bottleneck caused by clinical data characteristics such as heterogeneity and sparseness.

Corresponding author
Yi-Ju Tseng, yjtseng.info@gmail.com

## INTRODUCTION

On the basis of the development of electronic health records (EHRs), data analytics in health care is deemed as an essential process for accelerating the progress of clinical research (*Hersh, 2007*; *Jensen, Jensen & Brunak, 2012*; *Miotto & Weng, 2015*). Enriched EHRs contain crucial information related to disease progression, and this information can help with decision making in the health care field including for treatment selection and disease diagnosis (*Jensen, Jensen & Brunak, 2012*; *Raghupathi & Raghupathi, 2014*). However, processing and analyzing EHR data are usually challenging because of their heterogeneity and sparsity. These inherent characteristics create a common bottleneck in health care big data analytics (*Wu, Roy & Stewart, 2010*; *Hripcsak & Albers, 2013*; *Weiskopf & Weng, 2013*). Moreover, executing clinical data analysis project across different departments or institutes is difficult because clinical data formats and terminologies used to describe clinical conditions may vary across departments. A method that can standardize and facilitate the sharing of data or analysis pipelines from multiple sources is needed in research on clinical data analysis. Several common data models (CDMs) have been developed for eliminating clinical data format barriers, including the National Patient-Centered Clinical Research Network (PCORnet) (*Fleurence et al., 2014*; *PCORnet, 2020*) and Observational Medical Outcomes Partnership (OMOP) CDM (*Observational Health Data Sciences and Informatics, 2020*). The concept of CDM is to transform data into a CDM and terminology and then allow users to perform systematic analyses by using various sources. Although a CDM can help perform systematic analyses across different sources, the integration of clinical data and the preparation of analysis-ready data are unsolved issues.

The proposed open-source *dxpr* R package is a software tool aimed at expediting general EHR or claims data analyses through incorporating several functions that enable users to standardize, integrate, wrangle, and visualize clinical diagnosis and procedure records. Preparing an analysis-ready dataset from EHRs or claims data is a complex task that requires both medical knowledge and data science skills. The proposed *dxpr* package simplifies and accelerates the workflow for EHR data extraction and helps clinical data analysts generate simple and clean scripts that can easily be shared and reproduced. The *dxpr* package enables researchers to explore EHRs or claims data to acquire crucial information, understand disease progression, and analyze outcomes without writing complicated data preprocessing scripts. Moreover, the proposed package can support collaborative research across multiple data sources as long as the data include general diagnosis- or procedure-related information.

The *dxpr* package has three phases to process and analyze diagnosis codes in EHRs (Fig. 1). In the first phase, namely data integration, we transform diagnosis codes into a uniform format and provide four strategies to group diagnoses into clinically meaningful categories before the wrangling process. In the second phase, namely, data wrangling, users can use provided functions to query eligible cases, split data based on the index date, and calculate condition era according to the grouped diagnostic categories of each patients. Furthermore, exploratory data analysis preparation can be performed in this phase. Moreover, the *dxpr* package provides a function to convert a long format of grouped data into a wide format, which fits other analytical and plotting functions from

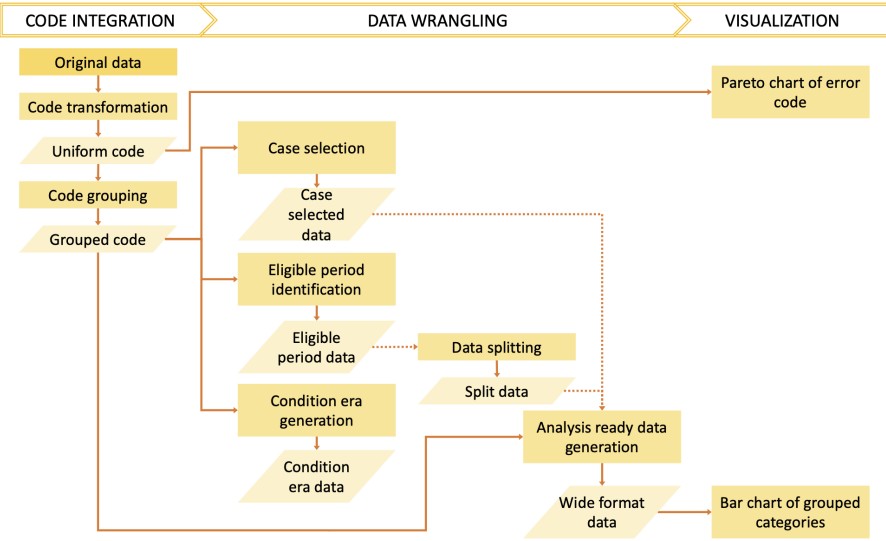

**Figure 1** Overview of the *dxpr* package.

other packages better. In the last phase, namely visualization, we provide overviews for diagnosis standardization and data integration, such as comorbidity distribution in the study population, comorbidity differences between case and control groups, and the most common diagnoses that failed to be grouped or transformed. The usage details are presented in the Supplementary Data S1 and S2. For processing and analyzing procedure codes, the concept is similar to diagnosis.

# MATERIALS AND METHODS

## Preparation

The current version of the package is available at Github (https://github.com/DHLab-TSENG/dxpr, Supplementary Data S3) and is accessible through the devtools package that enables installing packages from GitHub (*Wickham, Hester & Chang, 2020*). To install the *dxpr* R package, users can type the following commands in an R session:

```
devtools::install_github("DHLab-TSENG/dxpr")
library(dxpr)
```

The imported EHR dataset must contain at least three columns as indicated below:

- Member ID: a patient identifier, which can be numeric, alphanumeric, or a list of characters.
- Diagnosis/procedure code: ICD-9 or ICD-10 code assigned to a visit or an admission.

**Table 1  The first five diagnosis records of the sample dataset.**

| ID | ICD | Date |
|----|-----|------|
| A2 | Z992 | 2020-05-22 |
| A5 | Z992 | 2020-01-24 |
| A8 | Z992 | 2015-10-27 |
| A13 | Z992 | 2020-04-26 |
| A13 | Z992 | 2025-02-02 |

- Visit or admission date: the date of the visit, admission, or clinical service provided. The date should be in date format. If the date is recorded in a string format, it has to be recorded in year–month–day format (YYYY/MM/DD or YYYY-MM-DD).

Column names can be passed in each function by using function arguments.

The data can be imported from files or databases, with packages provide access to databases within R, such as DBI (*R Special Interest Group on Databases R-SIG-DB, Wickham & Müller, 2021*) and odbc (*Hester & Wickham, 2021*). We illustrate the use of the *dxpr* package with a diagnostic sample dataset of 10-year admissions of 38 patients, sampleDxFile, and the first five records are shown in Table 1.

## Data integration
### Code format transformation
The *dxpr* package first transforms ICD diagnostic codes into a uniform format before code grouping. ICD-9 and ICD-10 diagnostic codes (*US Centers for Medicare & Medicaid Services, 2017a*) have two formats, namely decimal (with a decimal place separating the code) and short formats. Different hospitals, grouping methods, or standards coded ICD into different formats. For example, studies using Clinical Classifications Software (CCS) (*HCUP, 2017*; *HCUP, 2019a*) and comorbidity measures, such as Elixhauser and Charlson (*Elixhauser et al., 1998*; *Menendez et al., 2014*; *Moore et al., 2017*), have coded the ICD in a short format, and a phenome-wide association study (PheWAS) (*Denny et al., 2010*) coded the ICD in a decimal format. Therefore, format transformation is required before code grouping, and the transformation type is decided by the chosen grouping method.

The transformation function (icdDxShortToDecimal) converts ICD-9 and ICD-10 codes into a uniform decimal format because a decimal format is needed for grouping diagnostic codes in PheWAS classification. Similar to icdDxShortToDecimal, icdDxDecimalToShort function converts diagnostic codes into a uniform short format, which can be used for grouping to CCS, Elixhauser, or other classifications. These transformative functions not only convert ICD codes into uniform format codes but also check for potential coding errors. We provide two types of warning messages: wrong ICD format and wrong ICD version. Additional suggestions are generated to help users adjust potential incorrect ICD codes if available.

```
ICD_Decimal <- icdDxShortToDecimal(dxDataFile = sampleDxFile,
                                   icdColName = ICD,
                                   dateColName = Date,
                                   icd10usingDate = "2015/10/01")

sampleDxFile$Decimal <- ICD_Decimal$ICD
head(sampleDxFile)
     ID  ICD       Date Decimal
1:   A2 Z992 2020-05-22  Z99.2
2:   A5 Z992 2020-01-24  Z99.2
3:   A8 Z992 2015-10-27  Z99.2
4:  A13 Z992 2020-04-26  Z99.2
5:  A13 Z992 2025-02-02  Z99.2
6:  A15 Z992 2023-05-12  Z99.2
tail(ICD_Decimal$Error)
       ICD count IcdVersionInFile    WrongType Suggestion
1:  75.52    4            ICD 9  Wrong format
2:  E03.0    4            ICD 9 Wrong version
3:    650    4           ICD 10 Wrong version
4: 123.45    3           ICD 10  Wrong format
5:  755.2    3            ICD 9  Wrong format      755.29
6:   7552    2            ICD 9  Wrong format       75529
```

### Code grouping

The code grouping functions collapse clinical diagnostic data (ICD-9/ICD-10 codes) (*US Centers for Medicare & Medicaid Services, 2017a*) into a smaller number of clinically meaningful categories that are more useful for presenting descriptive statistics than using individual diagnostic codes (*HCUP, 2019b*). The *dxpr* package supports four strategies to group EHR diagnosis codes, namely CCS (*US Centers for Medicare & Medicaid Services, 2017b*), PheWAS (*Denny et al., 2010*) (`icdDxToPheWAS`), comorbidity measures (*Elixhauser et al., 1998*; *Menendez et al., 2014*; *Moore et al., 2017*), and self-defining grouping methods. The CCS grouping strategies includes single-level CCS (`icdDxToCCS`) and multiple-level CCS (`icdDxToCCSLvl`) (*HCUP, 2017*; *HCUP, 2019a*), comorbidity measures (`icdDxToComorbid`) includes Elixhauser, Agency for Healthcare Research and Quality (AHRQ) and Charlson (*Elixhauser et al., 1998*; *Menendez et al., 2014*; *Moore et al., 2017*), and self-defining grouping methods includes precise matching (`icdDxToCustom`) and searching for lines containing a match (`icdDxToCustomGrep`). The grouping functions return two tables of the dataset, one is data with the corresponding grouping categories of each ICD (Table 2), and the other is summarized data exhibiting the earliest/latest record date and diagnosis counts in the same grouping category for each patient (Table 3). For example, after executing function `icdDxToCCS` for the records of patients A and B, two output types are shown in Tables 2 and 3, respectively. Patient A has three diagnosis records (ICD codes: 78550, 78552, and 785.59), which are all in the "shock" category of the CCS classification, with the earliest record on September 1, 2013 and the latest one on October 1, 2014. The `icdDxToCCS` function mapped corresponding CCS categories

**Table 2  Grouping results from grouping functions—icdDxToCCS.**

| Short | ID | ICD | Date | ccs_categories_description |
|-------|----|-----|------|---------------------------|
| 78550 | A | 78550 | 2014/10/01 | Shock |
| 78552 | A | 78552 | 2013/10/01 | Shock |
| 78559 | A | 785.59 | 2013/09/01 | Shock |
| 78552 | B | 78552 | 2013/09/01 | Shock |
| 25000 | B | 250.00 | 2012/07/01 | Diabetes mellitus without complication |
| 25000 | B | 250.00 | 2012/05/01 | Diabetes mellitus without complication |

**Table 3  Summarized results from grouping functions—icdDxToCCS.**

| ID | Categories | FirstCaseDate | EndCaseDate | Count | Period |
|----|------------|---------------|-------------|-------|--------|
| A | Shock | 2013/09/01 | 2014/10/01 | 3 | 395 days |
| B | Diabetes mellitus without complication | 2012/05/01 | 2012/07/01 | 2 | 62 days |
| B | Shock | 2013/09/01 | 2013/09/01 | 1 | 0 days |

for these ICD codes and returned the grouping results (Table 2). Similarly, patient B has two diagnosis records (ICD codes: 78552 and 250.00) in the "shock" category and "Diabetes mellitus without complication" category of CCS classification, and the grouping results are also shown in Table 2. According to these diagnosis records shown in Table 2, Table 3 shows that icdDxToCCS function can summarize the first and last dates of diagnosis, the total number of diagnoses, and the period between the first and last diagnoses for each category, which can be used for designing the analysis strategy. While icdDxToCCS groups codes into single-level CCS, icdDxToCCSLvl groups codes into multi-level CCS. Multi-level CCS expands single-level CCS into a four-level hierarchical system for diagnoses, which provide the opportunity to examine general aggregations or to assess specific conditions (*HCUP, 2019a*). For instance, if a user wishes to group codes into the second level of multi-level CCS, then this task can be performed through simply entering "ccslvl2" as the assigned grouping type. These grouping functions not only facilitate users to convert original diagnosis records from detailed levels into clinically meaningful diagnostic groups for further analysis but also provide aggregated information of each diagnostic group that can help research design and hypothesis generation, such as filtering out data based on specified criteria (e.g., first diagnosis dates of a specific chronic disease).

The usage of code classification function for CCS is as follows:

 

```
## ICD to CCS description
CCS_description <- icdDxToCCS(dxDataFile = sampleDxFile,
                                  idColName = ID,
                                 icdColName = ICD,
                                dateColName = Date,
                            icd10usingDate = "2015-10-01",
                              isDescription = TRUE)
CCS_description$groupedDT[CCS_description$groupedDT$ID=="A0",]
   Short ID   ICD       Date CCS_CATEGORY_DESCRIPTION
1:  5855 A0  5855 2013-12-20   Chronic kidney disease
2: V4511 A0 V4511 2012-04-05   Chronic kidney disease
3:  V560 A0  V560 2010-03-28   Chronic kidney disease
4:  5853 A0  5853 2010-10-29   Chronic kidney disease
5:  5856 A0  5856 2009-07-25   Chronic kidney disease
6:   001 A0   001 2014-11-05                     <NA>
7: A0.11 A0 A0.11 2017-01-31                     <NA>
8: A0.11 A0 A0.11 2023-08-12                     <NA>
head(CCS_description$summarised_groupedDT, 5)
    ID CCS_CATEGORY_DESCRIPTION firstCaseDate endCaseDate count    period
1:  A0   Chronic kidney disease    2009-07-25  2013-12-20     5 1609 days
2:  A1   Chronic kidney disease    2006-11-29  2014-09-24     5 2856 days
3: A10   Chronic kidney disease    2007-11-04  2012-07-30     5 1730 days
4: A11   Chronic kidney disease    2008-03-09  2011-09-03     5 1273 days
5: A12   Chronic kidney disease    2006-05-14  2015-06-29     5 3333 days
```

## Data wrangling
### Case selection
In clinical data analysis projects, the most crucial step is case definition and selection, such as defining Lyme disease cases from claims data (*Tseng et al., 2015*) or defining acute ischemic stroke from EHR (*Tseng et al., 2020*). The analysis results could change based on case definition and lead to a different conclusion. The query function `selectCases` can select cases matching case definitions. Users can select cases based on diagnosis (ICD) or diagnostic categories (CCS, PheWAS, comorbidities, or self-defined diagnostic categories). Moreover, the function provides an option to set the minimum number of diagnoses within a specific duration. For example, users can extract diabetes cases by assigning at least two diagnoses in ICD codes "250.xx" or "E10.x-E14.x" within 730 days when a user applies the validated diabetes case definition: "two physician claims within 2 years with diagnosis codes 250.xx or E10.x-E14.x" (*Chen et al., 2010*). The output dataset of this function provides the start and end dates of the cases, the number of days between them, and the most common ICD codes used in the case definition. Furthermore, a list of people who did not satisfy the required case conditions or practically match the case definition is appended in the returned output table, and these individuals can be defined as a control group or be removed.

```
Case <- selectCases(dxDataFile = sampleDxFile,
                        idColName = ID,
                       icdColName = ICD,
                      dateColName = Date,
                   icd10usingDate = "2015/10/01",
                     groupDataType = ccslvl2,
                     caseCondition = "Diseases of the urinary system",
                     isDescription = TRUE,
                         caseCount = 1,
                       periodRange = c(30, 365))
head(Case)
     ID selectedCase count firstCaseDate endCaseDate    period MostCommonICD MostCommonICDCount
1: A3      Selected     5   2008-07-08  2014-02-24 2057 days          V420                  3
2: A1      Selected     5   2006-11-29  2014-09-24 2856 days          5855                  2
3: A10     Selected     5   2007-11-04  2012-07-30 1730 days         V5631                  2
4: A12     Selected     5   2006-05-14  2015-06-29 3333 days          5859                  2
5: A13     Selected     5   2006-04-29  2025-02-02 6854 days          5855                  2
6: A15     Selected     5   2007-05-25  2023-05-12 5831 days         V5631                  2
tail(Case)
   ID selectedCase count firstCaseDate endCaseDate  period MostCommonICD MostCommonICDCount

1: D3 non-Selected    NA         <NA>        <NA> NA days          <NA>                 NA
2: D4 non-Selected    NA         <NA>        <NA> NA days          <NA>                 NA
3: D5 non-Selected    NA         <NA>        <NA> NA days          <NA>                 NA
4: D6 non-Selected    NA         <NA>        <NA> NA days          <NA>                 NA
5: D7 non-Selected    NA         <NA>        <NA> NA days          <NA>                 NA
6: D8 non-Selected    NA         <NA>        <NA> NA days          <NA>                 NA
```

### Eligible period identification

In some clinical data, such as claims data, individuals can join or leave the program on different dates, and the length of available records might affect the analysis completeness. The *dxpr* package provides a function `getEligiblePeriod` for researchers to identify the first/last record date for each patient. These outputs can be used as an index date for case exclusion, such as cases without at least 6 months washout or follow-up period, or further data splitting.

```
admissionDate <- getEligiblePeriod(dxDataFile = sampleDxFile,
                                    idColName = ID,
                                  dateColName = Date)
head(admissionDate)
     ID firstRecordDate endRecordDate
1:  D6       2005-10-09    2025-01-05
2: A12       2006-01-12    2022-06-12
3:  D1       2006-02-12    2024-04-04
4: A13       2006-04-29    2025-02-02
5:  A9       2006-06-30    2023-12-10
6:  D2       2006-09-01    2025-08-11
```

### Data splitting based on index date and moving window

In clinical data analysis projects, users usually need to extract data based on a specific clinical event (e.g., extracting data before the first Lyme disease diagnosis in the records (*Tseng et al., 2017*)). The date of the specific event (index date) can be the first/last record date of the events or patient record, and the table of the index date for each individual can be generated using `selectCases` or `getEligiblePeriod` function, respectively. The *dxpr* package provides a convenient function `splitDataByDate` that can split data through classifying the data recorded before or after the defined index date and calculating the period between the record date and index date based on a self-defined window. For example, if a user needs to aggregate the data by using a 30-day window, the data recorded on 15 and 45 days after the index date will be defined as window 1 and window 2, respectively. The output of `splitDataByDate` function helps users to split the data based on the study design, and this can be applied to further time-series multiple-measurement analysis with period information.

```
indexDateTable <- data.frame (ID = c("A0","B0","C0","D0"),
                              indexDate = c("2023-08-12", "2015-12-26",
                                            "2015-12-05", "2017-01-29"))
Data <- splitDataByDate(dxDataFile = sampleDxFile[grepl("A0|B0|C0|D0",ID),],
                        idColName = ID,
                        icdColName = ICD,
                        dateColName = Date,
                      indexDateFile = indexDateTable,
                              gap = 30)
Data[6:11,]
   ID   ICD      Date  indexDate timeTag window
1: A0   001 2014-11-05 2023-08-12      B    107
2: A0 A0.11 2017-01-31 2023-08-12      B     80
3: A0 A0.11 2023-08-12 2023-08-12      A      1
4: B0  N185 2015-12-26 2015-12-26      A      1
5: B0  N189 2017-11-27 2015-12-26      A     24
6: B0 A0.11 2017-12-19 2015-12-26      A     25
```

### Condition era generation

Condition era is a means to apply consistent rules for medical conditions to infer distinct episodes in care, generated through integrating distributed clinical records into a single progression record (*Reisinger et al., 2010*). The concept of condition era is committed to the length of the persistence gap: when the time interval of any two consecutive admissions for certain conditions is smaller than the length of the persistence gap, then these two admission events will be aggregated into the same condition era. Each condition era consists of one or many events, and differences between any two consecutive admission events are all within the persistence gap. For example, an episode of influenza may include single or multiple outpatient visits, and the length of the influenza course should be the period between the first and last visits of the episode. `getConditionEra` function calculates condition era by using the grouped categories or self-defining groups of each patient and then generates a table with individual IDs, the first and last record of an era, and the sequence number

of each episode. Users can easily convert scattered diagnoses into an episode of condition based on the chararistics of target disease progression with the proposed function.

```
Era <- getConditionEra(dxDataFile = sampleDxFile,
                       idColName = ID,
                       icdColName = ICD,
                       dateColName = Date,
                       icd10usingDate = "2015/10/01",
                       groupDataType = CCS,
                       isDescription = FALSE,
                       gapDate = 360)
head(Era)
   ID CCS_CATEGORY era firstCaseDate endCaseDate count   period
1: A0          158   1    2009-07-25  2010-10-29     3 461 days
2: A0          158   2    2012-04-05  2012-04-05     1   0 days
3: A0          158   3    2013-12-20  2013-12-20     1   0 days
4: A1          158   1    2006-11-29  2006-11-29     1   0 days
5: A1          158   2    2008-06-25  2008-06-25     1   0 days
6: A1          158   3    2012-06-19  2013-04-28     2 313 days
```

### Analysis-ready data generation

After data integration and wrangling, researchers often need to further analyze these processed data, and function `groupedDataLongToWide` converts the long format of grouped data into a wide format, which is fit for other analytical and plotting packages, such as tableone (*Yoshida & Bartel, 2020*) package.

```
CHARLSON <- icdDxToComorbid(dxDataFile = sampleDxFile,
                             idColName = ID,
                             icdColName = ICD,
                             dateColName = Date,
                             icd10usingDate = "2015-10-01",
                             comorbidMethod = CHARLSON)
groupedData_Wide <- groupedDataLongToWide(dxDataFile  = CHARLSON$groupedDT,
                                           idColName = ID,
                                           categoryColName = Comorbidity,
                                           dateColName = Date,
                                                 reDup = TRUE,
                                           numericOrBinary = B,
                                                 count = 2)
head(groupedData_Wide, 5)
   ID CANCER  CEVD  COPD DIAB_C  MSLD  PARA   PUD   PVD   RD Rheum
1  A0  FALSE FALSE FALSE  FALSE FALSE FALSE FALSE FALSE TRUE FALSE
2  A1  FALSE FALSE FALSE  FALSE FALSE FALSE FALSE FALSE TRUE FALSE
3 A10  FALSE FALSE FALSE  FALSE FALSE FALSE FALSE FALSE TRUE FALSE
4 A11  FALSE FALSE FALSE  FALSE FALSE FALSE FALSE FALSE TRUE FALSE
5 A12  FALSE FALSE FALSE  FALSE FALSE FALSE FALSE FALSE TRUE FALSE
```

## Visualization
### *Pareto chart of error ICD*

When code transformation is implemented in the *dxpr* package, it generates unified data of diagnosis codes with potential errors. Function `plotICDError` visualizes codes with potential error by using the Pareto chart containing a bar plot where error ICD codes are arranged in descending order, and the cumulative total is represented by the line. Users can sort based on the counts of error ICD codes and set the top selected number of the ordered dataset. For instance, if a user chooses the top 10 ordinal rankings, then the Pareto chart shows a plot of the top 10 common error ICD codes and a list with details of these 10 and other error ICD codes.

```
error <- icdDxDecimalToShort(dxDataFile = sampleDxFile,
                                icdColName = ICD,
                                dateColName = Date,
                              icd10usingDate = "2015/10/01")
plot1 <- plotICDError(errorFile = error$Error,
                          icdVersion = all,
                        wrongICDType = all,
                            groupICD = FALSE,
                              others = TRUE,
                                topN = 10)

plot1$ICD
         ICD count CumCountPerc IcdVersionInFile    WrongType Suggestion
 1:  A0.11    20       18.35%           ICD 10  Wrong format
 2:  V27.0    18       34.86%           ICD 10 Wrong version
 3:   E114     8        42.2%           ICD 10  Wrong format
 4: A01.05     8       49.54%            ICD 9 Wrong version
 5:  42761     7       55.96%           ICD 10 Wrong version
 6:  Z9.90     6       61.47%           ICD 10  Wrong format
 7:    F42     6       66.97%           ICD 10  Wrong format
 8:  V24.1     6       72.48%           ICD 10 Wrong version
 9:  A0105     5       77.06%            ICD 9 Wrong version
10:    001     5       81.65%            ICD 9  Wrong format       0019
11: others    20        100%            ICD 9  Wrong format
```

### *Bar chart of diagnostic categories*

Function `plotDiagCat` provides an overview of the grouping categories of the diagnoses and summarizes the proportion of individuals diagnosed with grouped diagnostic categories in the whole study population or case and control groups in a bar chart. Users can observe the number and percentage of diagnostic categories in their dataset through this function. Furthermore, this function compares the usage of significantly different diagnostic categories between case and control groups by using the chi-square test or Fisher's exact test when the data does not match the assumptions of the chi-square test. The default level of statistical significance is considered at 5% ($p = 0.05$). Researchers can set a threshold of the top N significant grouped categories and the minimum prevalence of the diagnostic groups in the case or control group.

The "`percentage`" column shows the proportion of individuals diagnosed with the diagnostic category in the group. For example, there are 38 patients in the sample file, and "Renal Failure" defined in Elixhauser comorbidity accounts for 63.16% of the population (24/38).

```
ELIX <- icdDxToComorbid(dxDataFile  = sampleDxFile,
                              idColName = ID,
                             icdColName = ICD,
                            dateColName = Date,
                        icd10usingDate = "2015-10-01",
                        comorbidMethod = ELIX)
groupedDataWide <- groupedDataLongToWide(dxDataFile  = ELIX$groupedDT,
                                              idColName = ID,
                                        categoryColName = Comorbidity,
                                            dateColName = Date,
                                                 reDup = TRUE,
                                        numericOrBinary = B)
plot2 <- plotDiagCat(groupedDataWide = groupedDataWide,
                             idColName = ID,
                                 topN = 10,
                            limitFreq = 0.01)

plot2$sigCate
    DiagnosticCategory  N Percentage
 1:           RENLFAIL 24     63.16%
 2:              TUMOR  6     15.79%
 3:               ARTH  5     13.16%
 4:              LYMPH  4     10.53%
 5:              PSYCH  4     10.53%
 6:               DRUG  3      7.89%
 7:              NEURO  3      7.89%
 8:               PARA  2      5.26%
 9:           PERIVASC  2      5.26%
10:              VALVE  2      5.26%
```

## Clinical procedure data processing

As diagnosis codes, ICD-9-Procedure Coding System (PCS) code also has two formats, namely decimal and short, whereas ICD-10-PCS code only has a short format. The functions (`icdPrToCCS` and `icdPrToProcedureClass`) provide two strategies (CCS and procedure class) to collapse ICD procedure codes into clinically meaningful categories for further analysis. This procedure has two CCS classifications: single and multiple levels. The usage is similar to the diagnostic CCS classification. A sample file (`samplePrFile`) is provided with procedure records, including three patients and 170 records.

The procedure classes (*HCUP, 2016*) are created to facilitate health services research on hospital procedures by using administrative data. The procedure classes provide a standard to categorize individual procedure codes into one of the four broad categories: minor diagnostic, minor therapeutic, major diagnostic, and major therapeutic. The aforementioned classification functions mentioned allow the researcher to readily determine whether a procedure is diagnostic or therapeutic and whether a procedure is minor or major in terms of invasiveness, resource use, or both.

## Use case

To illustrate the main features in the *dxpr* package and the typical workflow, we demonstrated an analysis using the package among newborns who were diagnosed with patent ductus arteriosus (PDA) from Medical Information Mart for Intensive Care-III (MIMIC-III) (*Johnson et al., 2016*). MMIC-III is a publicly available database comprising deidentified health-related data associated with the admissions of approximately 60,000 patients who stayed in the critical care units of the Beth Israel Deaconess Medical Center between 2001 and 2012.

We provided a sample file `sampleFile_MIMIC` obtained from MIMIC-III (*Johnson et al., 2016*), a medical dataset of 7,833 newborn patients with 45,674 admissions. This dataset is used for verifying the comorbidity difference between patients with and without PDA based on the *dxpr* package. In this example, we defined PDA cases as patients who had at least one PDA diagnosis (ICD-9-CM 7470*). The controls are defined as patients who never had PDA diagnosis.

## Performance analysis

The *dxpr* package is designated to accelerate the process of large EHR data integration and provide the ready-for-analysis dataset from the integrated EHR data. We verified the running time 100 times with a simulated dataset of 953,294 unique patients and 7,948,418 distinct diagnosis records in a standard personal computer with 64 GB DDR4 2133 GHz RAM and an Intel® Core™ i7-6700 (CPU @3.40 GHz), using Windows 10 (1809), R 4.0.1 (64 bits), and RStudio 1.2.5033.

# RESULT

## A use case—patients with PDA

We conducted comorbidity analyses based on a cohort of newborns from MIMIC-III ($n = 7,833$) by using *dxpr* and tableone (*Yoshida & Bartel, 2020*) packages. In the *dxpr* package, we first use `selectCases` function to define case (PDA) and control (non-PDA) groups. In total, 381 and 7,452 patients with and without PAD were included in our study, respectively. Then, `icdDxToComorbid` function was applied to group diagnoses into AHRQ-defined comorbidities. Finally, we analyzed and graphed the AHRQ-defined comorbidities based on `plot_groupedData` function (Fig. 2) by using the chi-square test and Fisher's exact test. To focus on comorbidities that were essential and recorded in adequate individuals in our study population, we excluded comorbidities recorded in <1% of the patients in the PDA or non-PDA group. The analysis-ready data generated by `groupedDataLongToWide` can be passed to the tableone (*Yoshida & Bartel, 2020*) package to create objects summarizing all comorbidities stratified by patients with and without PDA and by performing the statistical chi-square tests. The AHRQ comorbidity table revealed 8 of the 16 statistically significant comorbidities ($p < 0.05$, Table 4) among patients with and without PDA, and the comorbidities are visualized in Fig. 2.

## Performance

For a simulated dataset of 953,294 unique patients and 7,948,418 admission records, code grouping with CCS-defined comorbidities required $149 \pm 2.48$ s (including code

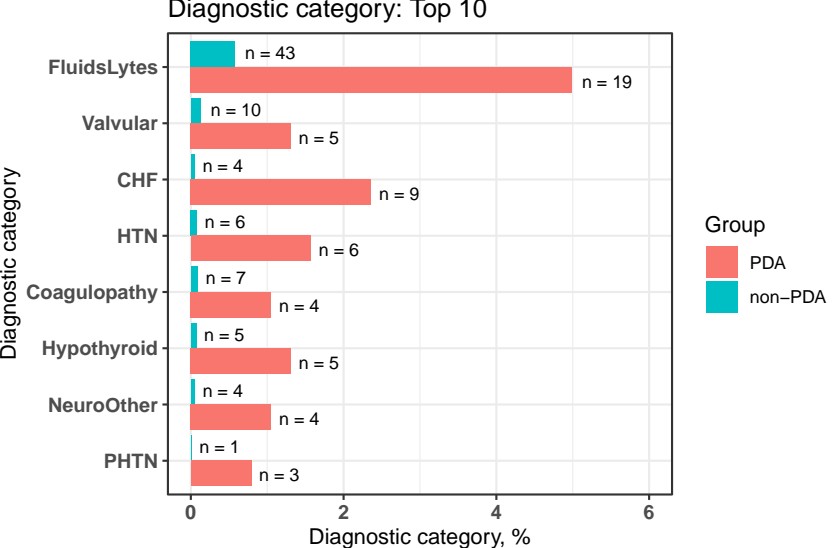

**Figure 2** Bar chart to visualize the statistically significant difference of diagnostic categories between patients with and without PDA, grouped by the AHRQ-defined comorbidities. PDA, patent ductus arteriosus; AHRQ, Agency for Healthcare Research and Quality; FluidsLytes, Fluid and electrolyte disorders; Valvular: valvular disease; CHF, congestive heart failure; HTN, hypertension, uncomplicated; Hypothyroid: hypothyroidism; NeuroOther, other neurological disorders; PHTN, pulmonary circulation disorders.

**Table 4** Summary of AHRQ-defined comorbidities based on the tableone package using the integrated data generated by the *dxpr* package.

| AHRQ[a] Comorbidities | Non-PDA | PDA[b] | p |
|---|---|---|---|
| *n* | 7452 | 381 | |
| Coagulopathy (%) | 7 (0.1) | 4 (1.0) | <0.001 |
| Congestive heart failure (%) | 4 (0.1) | 9 (2.4) | <0.001 |
| Deficiency anemias (%) | 2 (0.0) | 1 (0.3) | 0.342 |
| Depression (%) | 1 (0.0) | 0 (0.0) | 1 |
| Diabetes, complicated (%) | 2 (0.0) | 0 (0.0) | 1 |
| Fluid and electrolye disorders (%) | 43 (0.6) | 19 (5.0) | <0.001 |
| Hypertension, complicated (%) | 2 (0.0) | 0 (0.0) | 1 |
| Hypertension, uncomplicated (%) | 6 (0.1) | 6 (1.6) | <0.001 |
| Hypothyroidism (%) | 5 (0.1) | 5 (1.3) | <0.001 |
| Other neurological disorders (%) | 4 (0.1) | 4 (1.0) | <0.001 |
| Peripheral vascular disorders (%) | 1 (0.0) | 0 (0.0) | 1 |
| Pulmonary circulation disorders (%) | 1 (0.0) | 3 (0.8) | <0.001 |
| Renal failure (%) | 1 (0.0) | 0 (0.0) | 1 |
| Solid tumor without metastasis (%) | 1 (0.0) | 0 (0.0) | 1 |
| Valvular disease (%) | 10 (0.1) | 5 (1.3) | <0.001 |
| Weight loss (%) | 2 (0.0) | 0 (0.0) | 1 |

**Notes.**
[a] AHRQ: Agency for Healthcare Research and Quality.
[b] PDA: patent ductus arteriosus.

transformation). Case selection required $238 \pm 3.05$ s to query patients with diseases of the urinary system, eligible period identification required $1.12 \pm 0.22$ s to find the first and last admission date for each patient, data splitting with the first admission date for each patient required $6.50 \pm 0.42$ s, condition era generation required $372 \pm 6.39$ s, and analysis-ready data generation required $3.75 \pm 0.27$ s.

## DISCUSSION AND CONCLUSIONS

The *dxpr* package considerably simplifies the extraction, accelerates the processing of clinical data research, and enables researchers to prepare analysis-ready data with a standard workflow. The package had been developed and tested using structured clinical data, such as critical care data (MIMIC-III (*Johnson et al., 2016*)), a multi-institutional medical care database (Chang Gung Research Database (*Tsai et al., 2017*; *Tseng et al., 2020*)), and claims data (National Health Insurance Research Database (*Hsieh et al., 2019*)), indicating that the package can be applied to data from different countries, institutions, and data structures. The available functions are summarized in Table 5.

Several software and packages were developed to facilitate clinical data analysis. rEHR (*Springate et al., 2017*) established a clinical data analysis workflow to simplify the processing of EHR. The rEHR package simplifies the process of extracting data from EHR databases. It used the database backend that can accelerate data access and process times. However, this design needs database backend, which might not be suitable in many circumstances. Furthermore, the international diagnosis coding standard, such as ICD, were not used in the package. The ICD (*Wasey & Lang, 2020*) package is designed for calculating comorbidities and medical risk scores with ICD-9 and ICD-10 codes. It is helpful to group ICD codes according to comorbidities. However, in clinical data analysis, eligible case selection, data split based on the defined index date, and visualization are also essential. Therefore, we designed and developed the *dxpr* package to facilitate diagnosis data analysis.

The proposed package has limitations, which come from either the data or package itself. For analyzing clinical data, the *dxpr* package highly depends on diagnosis and procedure codes, but these codes may vary in accuracy across different institutions. Furthermore, the effect of switching diagnosis codes from ICD-9 to ICD-10 should be considered if the analysis period is across the switching date. In addition to diagnosis and procedure data, the other data not included in proposed packages, such as medication data, are important in clinical data analysis projects. In the R ecosystem, the AdhereR (*Dima & Dediu, 2017*) package implements a set of functions that are consistent with current adherence guidelines and definitions. Fourth, we provide an easy-to-use package that will help analysts process raw data and notify them when potential coding errors exist. However, even with this package, analysts should understand their data precisely. This easy-to-use package will help analysts process clinical data with its coding error–checking functions, but may also lead naïve analysts to miss opportunities to find other errors in the data. Finally, the *dxpr* package is focused on analysis-ready data generation so that the statistic method incorporation may be insufficient. However, the R ecosystem's most significant advantage is that many

**Table 5  Functions in the *dxpr* package.**

| Functions | Descriptions |
|---|---|
| **I. Data integration** | |
| icdDxShortToDecimal | Transform ICD[a] diagnostic codes into decimal format |
| icdDxDecimalToShort | Transform ICD diagnostic codes into short format |
| icdDxToCCS | Group ICD diagnostic codes into single CCS[b] category |
| icdDxToCCSLvl | Group ICD diagnostic codes into multiple CCS category |
| icdDxToComorbid | Group ICD diagnostic codes into comorbidity category (Elixhauser, Charlson, and AHRQ) |
| icdDxToPheWAS | Group ICD diagnostic codes into PheWAS[c] category |
| icdDxToCustom | Group ICD diagnostic codes into customized grouping category based on precise method |
| icdDxToCustomGrep | Group ICD diagnostic codes into customized grouping category based on fuzzy method |
| **II. Data Wrangling** | |
| selectCases | Query matching cases in the EHR[d] data |
| splitDataByDate | Query data by a clinical event |
| patientRecordDate | Query the earliest/latest admission date for each patient. |
| getConditionEra | Calculate condition era by grouped categories of each patient. |
| groupedDataLongToWide | Convert long format of grouped data into wide format for analytical and plotting functions |
| **III. Visualization** | |
| plotICDError | Pareto chart of error ICD list |
| plotDiagCat | Bar chart of diagnostic categories |
| **Procedure** | |
| icdPrToCCS | Group ICD procedure codes into single CCS category |
| icdPrToCCSLvl | Group ICD procedure codes into multiple CCS category |
| icdPrToProcedureClass | Group ICD procedure codes into procedure class category |

**Notes.**
[a] ICD, International Classification of Diseases.
[b] CCS, Clinical Classifications Software.
[c] PheWAS, Phenome Wide Association Studies.
[d] EHR, Electronic Health Record.

well-developed packages were developed to facilitate statistical analysis. In the use case demonstration, our package can be used with other packages, such as tableone package. The tableone (*Yoshida & Bartel, 2020*) package is developed to ease the construction of the common "Table 1" in research papers, providing patient baseline characteristics table with summary statistics and hypothesis tests.

We demonstrated that the *dxpr* package can play an essential role in complex clinical data preprocessing and analysis-ready data generation through integrating the international standard of clinical data. This package helps clinical data analysts combat the common bottleneck caused by certain clinical data characteristics, such as heterogeneity and sparseness.

## ACKNOWLEDGEMENTS

We thank Ru-Fang Hu, Yi-An Zhu, and Chia-Wei Chang, from Department of Information Management, Chang Gung University, for testing the *dxpr* package. This manuscript was edited by Wallace Academic Editing.

### Funding

This research was supported by Chang Gung Memorial Hospital (CMRPD3K0011), the Ministry of Science and Technology, Taiwan (MOST 109-2636-E-008-009 and MOST 110-2636-E-008-008). There was no additional external funding received for this study. The funders had no role in study design, data collection and analysis, decision to publish, or preparation of the manuscript.

### Grant Disclosures

The following grant information was disclosed by the authors:
Chang Gung Memorial Hospital: CMRPD3K0011.
Ministry of Science and Technology, Taiwan: MOST 109-2636-E-008-009,  MOST 110-2636-E-008-008.

### Competing Interests

The authors declare there are no competing interests.

### Author Contributions

- Yi-Ju Tseng conceived and designed the experiments, performed the experiments, analyzed the data, performed the computation work, authored or reviewed drafts of the paper, and approved the final draft.
- Hsiang-Ju Chiu and Chun Ju Chen conceived and designed the experiments, analyzed the data, performed the computation work, prepared figures and/or tables, authored or reviewed drafts of the paper, and approved the final draft.

### Data Availability

The sample code of the dxpr package is available at GitHub:

https://github.com/DHLab-TSENG/dxpr-paper/blob/main/SampleCode.md

A detailed example of the usage of the dxpr package is available at GitHub:

https://github.com/DHLab-TSENG/dxpr-paper/blob/main/Supplement.md

The source code of the dxpr package is available at GitHub:

https://github.com/DHLab-TSENG/dxpr.

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
