# Peer review of "dxpr: an R package for generating analysis-ready data from electronic health records—diagnoses and procedures"

_PeerJ Computer Science, doi:10.7717/peerj-cs.520_

## Round 0.1 · original submission · Major Revisions

Technically speaking, this paper is well-written to design and develop a package for the raw data. At the same time, the motivation is not clearly following the comments from reviewer 1. I strongly recommend the authors to re-organise the contents and highlight the practical application in clinic. It implies that a major revision is needed at least.

·

Basic reporting

See General Comments

Experimental design

See General Comments

Validity of the findings

See General Comments

Additional comments

This well-written though improperly formatted and presented article proposes a common data model (CDM) R algorithm, dxpr, that would address analysis “bottlenecks” inherent in clinical data sets due to “heterogeneity and sparseness” through pre-processing to output analysis-ready data sets. I confess that, for the most part, this is not a challenge that I have to deal with as I am fortunate to work for an agency (DoD) that has consistently formatted medical data. However, I have often dealt with those and other issues when synthesizing data from diverse sources.

I do not work with R, but exclusively with SAS, so I make no pretense of critiquing the coding per se. The application, as described, appears to be flexible, i.e. it provides the option for CDM categorizations (CCS, Elixhauser, Charlson comorbidities, etc.) as well as user-defined. The standardization, in terms of formatting of ICD dx and procedure codes is a welcome essential feature.

I would not use this product--partly because I don’t work with R although I see no reason why my raw data couldn’t be pre-processed in dxpr and then analyzed using SAS. I also would not make use of the point-and-click menu analysis features in SAS Enterprise Guide (I generally despise SAS EG). The fact is having an intimate understanding of the raw data is essential for executing any competent statistical analysis. I would not consider delegating that responsibility, or rather, that opportunity. Occasionally, a well-meaning potential customer, who cannot access raw data, will offer to perform their own analysis. “All I need you to do is clean, manipulate, and synthesize the data.” Just that. They’re happy to do the high level work of pressing the regression button. If there is a perceived analysis bottleneck I suspect it is among practitioners who perform analysis as a secondary function.

I am instinctively uneasy with the notion of producing “analysis-ready” data. That said, we employ metrics, imperfect though they all are, to find common ground for comparisons. Knowing that other investigative groups operating under different health care systems (perhaps in different countries) employed the same algorithm to clean and prepare their data would seem to provide a strong starting point for comparison. Nevertheless, analysis bottlenecks are not entirely a bad thing. Making complex clinical data more accessible sounds like a good idea but lowering the requirements for entry may permit access to naïve analysts. Ultimately though, the onus is on the investigator(s) to competently execute their analysis. Dxpr seems like a useful product—if used responsibly. But that is always the caveat.


It’s not clear why this dxpr package description was presented as an analysis—which it isn’t. I don’t care for the arbitrary case-control “analysis” described in the results. (Selecting controls from a sample of sick people is not best practice and at any rate, if dxpr is about creating analysis ready data a contrived example of an analysis is not necessary.) There was no mention of it in the Methods. If the aim is to describe the dxpr package then please don’t pretend to do something else. One would expect the editorial staff to address these issues before sending the article out for review.

Recommendation: Major revision/re-formatting

I have no interest in reviewing a second time.

Reviewer 2 ·

Basic reporting

The paper is well-written and easy to read and comprehend

Experimental design

no comment

Validity of the findings

The paper introduces an R packages that seems to do what is supposed to do, but this is relatively hard to asses in detail. That being said, I wold have expected more unit tests (now there seems to be just one very general such test)

Additional comments

I found the paper well written and I think the package is a useful addition to the computational ecosystem.
However, I do have some comments, questions and suggestions:

abstract: "combat" -> "address"?
p9, 100-107: this text seems duplicated from the intro and does not really seem to be about methods...
p10 112-113: please give a better ref to the devtools package
p10 123: so, the date format cannot be changed? why?
p16 305-307: the index date must be pre-computed, as it seems to be a fixed date -- is this correct? I do see why this choice but I was wondering if a different approach might be useful as well (say, x days since the first event) or so?
condition eras: the gapDate is fixed for all patients and conditions, right? why?

The code: I looked at the GitHub repo and I do have some general questions:
1. (touched above): I feel that a good set of unit tests are essential, especially as the package will continue to be developed
2. help: I found that the help could be improved and that, in some cases, not info is given about the expected parameters (e.g., "dxDataFile" is actually *not* a file, but a data.frame-like object but this is not specified)
3. the I/O seems limited to in-memory only (and I fully understand that) but it would be useful to see a discussion (maybe with some code) about how to use the package with, say, and SQL database?

---

## Round 0.2 · Minor Revisions

Based on the reviewer's comments, a minor revision is still needed for the current version, while the structure and format of the paper should be updated and the contribution is believed to be acceptable.

Reviewer 2 ·

Basic reporting

no comment

Experimental design

no comment

Validity of the findings

no comment

Additional comments

Thanks for revising the paper and taking my comments seriously. However, there are a few (minor ones) that still stand and that would improve the paper. I will re-mention them here:

- please give a better ref to the devtools package: what I meant is to give a proper reference to the paper in the academic sense. Running `citation("devtools")`
gives:
@Manual{,
title = {devtools: Tools to Make Developing R Packages Easier},
author = {Hadley Wickham and Jim Hester and Winston Chang},
year = {2020},
note = {R package version 2.3.2},
url = {https://CRAN.R-project.org/package=devtools},
}
which should be used instead of the current "(Wickham H)"

- so, the date format cannot be changed? why?: now the documentation seems to specify that "As for date column, the data format should be YYYY/MM/DD or YYYY-MM-DD" -- I still wonder why not letting the user specify the date format (see, for example, the `lubridate` package)

- gapDate: ok, thanks for the clarification; while I agree that it makes sense in most cases to fix it per condition, I am wondering if there might be particular cases where you still want to let vary *also* between patients?

- unit tests: what I meant is the stuff that resides in the `dxpr/tests/testthat` folder, which now is just a skeleton `test-icdToCCS.R`; it is always a good idea to have a battery of unit tests to make sure that evolutions/bug fixes do not break things (e.g., https://towardsdatascience.com/unit-testing-in-r-68ab9cc8d211)

Otherwise, thanks for updating the documentation and the vignettes, including with accessing SQL data.

---

## Round 0.3 · accepted · Accept

After the major and minor revision, the manuscript has been improved regarding contribution and quality. All the concerns from the reviewer have been addressed well in the revised version and now I believe that the current version meets the criteria for publishing. I recommend to accept it as it is. Congratulations and looking forward to the final edited version.

Reviewer 2 ·

Basic reporting

no comment

Experimental design

no comment

Validity of the findings

no comment

Additional comments

Thanks for addressing all my previous comments!